# Effective Tensile Strength Estimation of Natural Fibers through Micromechanical Models: The Case of Henequen Fiber Reinforced-PP Composites

**DOI:** 10.3390/polym14224890

**Published:** 2022-11-12

**Authors:** Francesc Xavier Espinach, Fernando Julian, Manel Alcalà, Fabiola Vilaseca, Félix Carrasco, Pere Mutjé

**Affiliations:** 1LEPAMAP-PRODIS Research Group, University of Girona, Maria Aurèlia Capmany 61, 17003 Girona, Spain; 2Advanced Biomaterials and Nanotechnology, University of Girona, Maria Aurèlia Capmany 61, 17003 Girona, Spain; 3Department of Chemical Engineering, University of Girona, Maria Aurèlia Capmany 61, 17003 Girona, Spain

**Keywords:** henequen, polypropylene, composites, tensile strength, micromechanical analysis

## Abstract

The performance of henequen fibers and polypropylene composites obtained by injection molding with and without coupling agent was evaluated. Henequen fibers are natural non-wood fibers mainly used in textile sector or in thermosetting matrix composites. In this work, henequen fibers have been used as a possible substitute reinforcement material for sized glass fibers. The surface charge density of the materials used was evaluated, as well as the morphology of the fibers inside the material. A significant reduction in the length of the fibers was observed as a consequence of the processing. The use of a 4% coupling agent based on fiber content was found to be effective in achieving significant improvements in the tensile strength of the composites in the reinforcement range studied. The influence of the aspect ratio on the coupling factor was determined, as well as the evaluation of the interface quality. The results obtained demonstrate the great potential of henequen fibers as reinforcement of composite materials, giving rise to strong interfaces with coupling. Finally, the comparison of henequen fiber composites with sized glass fiber composites showed that it is possible to substitute polypropylene composites with 20 wt.% glass fiber for 50 wt.% henequen fibers.

## 1. Introduction

Henequen fiber is a natural non-wood fiber extracted from the Agave plant. This type of crop is native to Mexico, specifically the Yucatan Peninsula. Henequen fibers are extracted from the leaves of the plant that are more than 90 cm long. The fibers obtained are long, hard and strong fibers, with properties similar to sisal fibers. The chemical composition of henequen fibers is characterized by a low presence of lignin, being approximately 68% cellulose, 18% hemicellulose, 9% lignin, 4% extractives, and 1% ash [1,2]. Industrially, henequen fibers are commonly used in the textile sector for the manufacture of twine, rope, carpets, and cordages [2]. As a result of these applications, the production of henequen fibers has increased significantly, which has led to its use as a reinforcement for plastic materials and cement [3,4,5].

The main limitation of natural fibers is their thermal resistance and therefore limitations in the processability of thermoplastics transformation. It is widely known that the processing temperature should be below 200 °C with fast processing times to avoid their degradation [6]. In this sense, most of the studies carried out with henequen fibers have focused on their use in thermosetting plastic matrices. The use of thermosetting resins is common in the production of composite materials from natural fibers due to their easy processability and greater facility to obtain good yields, unsaturated polyester resins being the most used thermosetting matrices [7,8,9]. However, the development of composite materials with short natural fibers in thermoplastic matrices has experienced an important boom in the last few years [10,11,12,13]. Although there is a major problem with plastic materials worldwide, the complete substitution of polymers as widespread as polypropylene is not possible today. One of the main compounds that can be found on the market are those made of polypropylene with glass fiber. These compounds present a double problem since their origin for both the matrix and the reinforcement is of fossil origin and the use of fiberglass represents a significant health risk for the workers who handle it [10,14,15,16]. It is for this reason that the incorporation of natural fibers as substitutes for glass fibers in matrices such as polypropylene is already a great technological advance, giving rise to materials with a high percentage of bio-based material, recyclable, durable and without risk to the health of workers [10,17,18]. However, achieving natural fiber composites with considerable improvements in mechanical properties is not simple. The different surface polarity between the plastic matrix and natural fibers with a high content of hydroxyl groups hinders the creation of a strong interface [19]. It is known that a strong bonding in the interfacial region (fiber–matrix), as well as a correct dispersion of the fibers inside the material, are key parameters in achieving the high mechanical performance of a composite material. There are many studies where researchers have focused on the surface modification of fibers by chemical modifications, such as alkalization, acetylation, and silane coupling [20,21,22] or physical modifications with plasma [23]. However, as already found in the previously published work, achieving improvements in the mechanical properties of henequen fiber-reinforced polypropylene composites can be easily achieved by incorporating the appropriate coupling agent content [1].

This study evaluates the tensile properties of polypropylene composite materials reinforced with henequen fibers, focusing on the evaluation of the interface strength.

This study evaluates the tensile properties of polypropylene composites reinforced with henequen fibers, focusing on the evaluation of the interface strength. The influence of the coupling agent, optimized in previous works, on different percentages of henequen fibers was investigated. Polypropylene composite materials with sized glass fiber have been produced for comparison, to determine the substitution capacity that henequen fibers can offer. Significant improvements in tensile strength have been observed when incorporating the coupling agent. In this sense, the Fiber Tensile Strength Factor (FTSF) parameter used as a basis for the comparison of the materials was initially evaluated. Then, the quality of the interface was studied through micromechanical models, by determining the coupling factor, the orientation factor, the length and interface factor, and the shear strength. Likewise, the intrinsic resistance of the henequen fibers inside the material was mathematically determined.

## 2. Materials and Methods

### 2.1. Materials

The matrix used to obtain the composite materials was homopolymer polypropylene (PP) of ISPLEN PP090 G2M grade, which was kindly supplied by Repsol Química, S.A. (Tarragona, Spain). The PP used has a Melt Flow Rate (MRF) of 30 g/10 min at 230 °C and a weight of 2.16 kg, and its density is 0.950 g/cm^3^. The coupling agent used was maleic acid graphitized polypropylene (MAPP) of Epolene G3015 grade with an acid number of 15 mg KOH/g, and a molecular weight of 24800 Da. The MAPP was supplied by Eastman Chemical Products (San Roque, Spain). Henequen fibers used as reinforcement were produced and supplied by Centro de Investigación Científica de Yucatán (CICY) (Mérida, Mexico) from agave (Agave fourcroydes). The sized glass fibers (GF) used as comparative reinforcement were produced by Vetrotex (Chambéry Cedex, France) and supplied by Maben SL (Banyoles, Spain). The decahydronaphthalene (decalin) used for the extraction of the fibers from the composites produced was supplied by Fisher Scientific (Hampton, New Hampshire, USA). The methyl-glycol-chitosan (MGCh) used for surface charge density evaluation was supplied by Wako Chemical GMBH (Neuss, Germany). The other chemical reagents used in the different processes, such as acetone, were purchased from Merck, KGaA (Dramstadt, Germany).

### 2.2. Methods

Figure 1 shows the flow diagram of the procedure carried out in this study.

Before any processing step, both the polymeric matrix, the coupling agent, and the fibers are dried in an oven at 80 °C for at least 48 h to avoid the presence of moisture.

The surface charge density of fibers and polymers were determined by titration of a finely powdered suspension with MGCh as a cationic reagent. MGCh interacts with the polar groups of the fibers or polymer surfaces [24]. MGCh was added in excess, and after interaction of MGCh and polar groups, the excess that did not interact was titrated with a potassium polyvinyl sulfate solution and using o-toluidine (TBO) blue dye as an indicator.

The incorporation of the fibers, in both cases Henequen fibers and glass fibers sized, was carried out by mixing them in a Brabender Plastograph^TM^ (Alton, United Kingdom) at 180 °C and 80 rpm. The process starts by incorporating the polypropylene matrix or the polypropylene matrix according to the experiment performed with and without a coupling agent. Once the plastic polymer reaches the melting temperature, the desired amount of fiber is incorporated and held for 8 min for compounding. Once the compounding time is over, the material obtained is cooled to room temperature for subsequent pelletizing using a Retsch^TM^ SM 100 pelletizer (Düsseldorf, Germany). The composites obtained with henequen fibers have been limited to a weight percentage of 50%, given the impossibility of polypropylene to incorporate a higher amount of fiber and maintain its cocontinuity. The incorporation in the Brabender equipment of higher contents resulted in a discontinuous material with unattached fibers, impossible to process.

The material obtained is again kept at 80 °C before processing. The transformation of the composite material obtained was carried out by injection molding. For this purpose, a mold with the shape of the standard specimens was used to characterize the tensile properties according to ASTM D638. The transformation of the composite was carried out in a Meteor-40 injection-molding machine of Mateu & Solé (Barcelona, Spain). This injection-molding machine has three heating zones, which were set at 175 °C, 175 °C, and 190 °C, the highest temperature being that of the nozzle. The first and second pressures used for injection were 120 and 37.5 bars, respectively.

The standard specimens obtained were then conditioned according to ASTM D638 in a Dycometal climatic chamber (Viladecans, Spain) at 23 °C, 50% relative humidity for at least 48 h before evaluation. The tensile properties of the specimens obtained were tested on an Instron^TM^ 1122 universal testing machine equipped with a 5 kN load cell. The testing speed was 2 mm/min. As is well known, the use of natural fibers with high disparity and presence of defects can lead to high deviations in the results. Therefore, all tests have been performed a minimum of 10 times to obtain the arithmetic mean and standard deviation presented in tables and graphs.

Part of the specimens obtained were used to study the morphology of the fibers inside. For this it was necessary to perform a Soxhlet extraction where the polymeric matrix was solubilized using decalin as solvent. The specimens were cut into smaller pieces and placed inside a cellulose cartridge of the dimensions of the Soxhlet equipment together with a piece of cotton used to prevent the pieces from coming out of the cartridge during reflux. The extraction was carried out at reflux for 48 h. At the end of the extraction time the fibers present in the cellulose cartridge were washed repeatedly with acetone and finally with distilled water. Finally, the fibers obtained were characterized by morphological analysis using the Morfi fiber analyzer of Techpap (Gières, France). Morphological analysis was used to obtain the distribution of fibers, as well as the average values of arithmetic length, weight-weighted length, and diameter.

The qualitative observation of the fiber–matrix interface was performed by microscopic analysis of the fracture-surface area. This observation was performed by scanning electron microscopy (SEM) in a Zeiss DMS 960 (Oberkochen, Germany) at different magnifications.

## 3. Results and Discussion

For the evaluation of the tensile properties of composite materials, Young’s modulus tensile strength and deformation capacity of the material should be considered separately. As has been widely reported in the literature, the tensile strength properties of composite material are the combination of the properties of the polymer matrix and fibers, as well as the strength of the interface [1,25,26,27]. This is not the case for Young’s modulus where it is not dependent on the interface [28,29,30].

### 3.1. Macromechanical Analysis of Tensile Properties

Equation (1) presents the modified rule of mixtures, where the different factors that influence the tensile strength of the composite material are defined. This equation divides the tensile strength of the composite into the contribution of the polymer matrix ($\left(1-V^{F}\right)\cdot \sigma_{t}^{m^{*}}$) and the contribution of the fiber ($\sigma_{t}^{F}\cdot V^{F}$) multiplied by the efficiency factor or coupling factor ($f_{c}$).
(1)$$\sigma_{t}^{C}=f_{c}\cdot \sigma_{t}^{F}\cdot V^{F}+\left(1-V^{F}\right)\;\cdot \;\sigma_{t}^{m^{*}},$$

The influence of the properties of the polymeric matrix is not only limited to its tensile strength; its density, melt flow rate, Young’s modulus, and deformation capacity plays an important role when incorporating fibers [31,32,33]. The most relevant characteristics of the fibers are the fiber type, the percentage of reinforcement, the dispersion and orientation of the fibers within the matrix, the aspect ratio (length divided by diameter), and the interfacial shear strength [27,34,35].

To determine the contribution of the matrix to the composite strength, $\sigma_{t}^{m^{*}}$ value is calculated. This value is obtained from the elongation at break of the composite material and the stress-strain correlation of the polymer matrix (Equation (2)). Thus, it is considered that the contribution of the matrix to the strength of the material can never be higher than the contribution of the matrix to the strength of the material at that level of deformation.
(2)$$\sigma_{t}^{m}=0.003\varepsilon^{5}-0.0751\varepsilon^{4\;}+0.757\varepsilon^{3}-4.0768\varepsilon^{2}+13.966\varepsilon +0.083,$$

The tensile strength results for composites with and without coupling agent are presented below in Table 1. The amount of 4% coupling agent used was optimized in previous work published by Tarrés et al., 2019 [1].

If the contribution of henequen fibers to increase the tensile strength of the composite material without and with a coupling agent is analyzed at the micromechanical level, it can be observed that there is a significant difference (Table 1). Although it is possible to achieve a certain reinforcement of the material by incorporating percentages higher than 10 wt.% by weight of henequen fiber, these are clearly limited by the fiber–matrix interface. The maximum increase was achieved with 50 wt.% reinforcement is only 38.4%. Values similar to those of other natural fibers without coupling agent [36]. If we consider that the composite material reinforced with 50 wt.% henequen has a density of 1090 g/cm^3^, compared to the density of polypropylene 0.905 g/cm^3^, the specific tensile strength only increases by approximately 15% (from 30.5 to 35.1 kN-m/kg). On the contrary, by incorporating 4% MAPP as a coupling agent we can observe how the increase in tensile strength for the same percentage of reinforcement is much higher. Comparing again the 50 wt.% of reinforcement, we can see how the increase in tensile strength is 89.9%. Similarly, if we consider the increase in the density of the material caused by the presence of the henequen fibers, we can observe an increase in the specific tensile strength of approximately 58% (from 30.5 to 48.1 kN-m/kg) in the composite with 50 wt.% reinforcement. Additionally, as reported in the literature, a linear increase in composite tensile strength as a function of reinforcement content is indicative of a strong interface, whereas non-linearity in the increase in tensile strength can be attributed to a weak interface [1,37]. The results obtained show henequen fibers as fibers with high reinforcement potential when a strong interface is obtained. Composites with 30 wt.% henequen allowed achieving an increase of 11.23% and 47.83% in tensile strength without and with coupling agent, respectively. In the literature, we can find works with other natural fibers where with 30 wt.% reinforcement without coupling agent similar increases are achieved [38]. On the other hand, Bhagat A. and Ghosh A. 2022 reported similar increases in tensile strength, approximately 46%, when using sisal fibers and polypropylene graphitized with maleic anhydride [39]. In this same work, the authors demonstrated the ability of maleic anhydride to create covalent bonds and hydrogen bonds with the hydroxyl groups of natural fibers, resulting in a strong interface [39].

In Figure 2, electron microscopy images of the composites with and without coupling agent are shown to analyze the fiber–matrix interface.

Microscopy images confirm the weak interface of the composites without coupling agents, presenting holes in the section of the material where the fracture has taken place (Figure 2d–f). The presence of these holes confirms the absence of a strong interface, resulting in fiber slippage when the material is stressed. On the contrary, in the images obtained from composites with 4% MAPP, it is possible to observe fiber breakage (Figure 2a). Likewise, in Figure 2b,c it is possible to observe the anchorage of the fiber in the continuous phase of the composite material (matrix). Therefore, the increase in the tensile strength of the composite materials with a coupling agent is caused by a strong fiber–matrix interface and therefore a greater transmission of stresses [40,41,42,43]. These results agree with those presented by Poornima et al., 2022 [44]. By scanning electron microscopy images, the influence of the coupling agent on basalt and polypropylene fiber composites was shown. In this work, it was shown that by the addition of a compatibilizer the fibers were uniformly dispersed and bonded, while without the presence of this coupling agent, voids or holes were detected due to the displacement of the fibers caused by the application of force [44].

It is widely known that the difference in polarity between plastic materials and natural fibers makes it difficult to create strong interfaces [19,45]. In this sense, Table 2 presents the results of the surface charge density test with MGCh.

As shown in Table 2, the surface charge density, expressed as micro equivalents of MGCh per gram of material, is much higher for natural fibers than for plastic polymers [46]. Similarly, if we compare the surface charge density of Henequen fibers with other similar fibers, these are in the highest range, together with Sisal fibers (above 20 µeq/g). The higher surface charge density of henequen fibers to bleached pulps, traditionally used for paper production, may be due to the higher presence of lignin and extractives [19,47].

The higher polarity of natural fibers is often associated with the creation of fiber aggregates. The formation of these aggregates hinders the dispersion of the fibers in the matrix. However, the evolution of Young’s modulus (Table 1) shows a linear evolution in both composites with and without a coupling agent. It is known that Young’s modulus is not affected by the interface strength and that it varies according to the intrinsic properties of matrix and reinforcement, being its behavior linear as the reinforcement content increases when a correct dispersion of the reinforcement is obtained. On the other hand, these high differences in polarity, 20.67 µeq per gram of fiber versus 4.56 µeq per gram of polypropylene, hinder the formation of a strong interface. It is known that high differences in surface charge density hinder the transfer of stress between the fiber and the matrix [47]. Therefore, the presence of a coupling agent with a structure similar to that of the matrix but with the ability to chemically bond to the fiber is essential to improve the strength of the interface. As shown in Table 2, the coupling agent has a surface charge density similar to that of PP. The slight difference observed may be due to the presence of graphitized maleic anhydride groups on the polypropylene surface.

The presence of maleic anhydride groups promotes the bonding between MAPP and natural fibers through covalent bonds and hydrogen bridges as reported in previous work [1,48]. However, the presence of the coupling agent does not only have effects on tensile strength by improving the interface. Its presence also modifies the deformation capacity of the material by providing greater elongation at break due to the anchorage of the fibers in the matrix. In the case of 40 wt.% reinforcement, the elongation at break was 3.3% compared to 1.9% for the composite without a coupling agent. This higher elongation at the breaking point of the material in turn allows a higher contribution of the matrix in the final strength of the composite expressed by the $\sigma_{t}^{m^{*}}$ value in the case of a 40 wt.% of 21.2 MPa compared to 16.2 MPa of the composite without a coupling agent. Therefore, it is found that the presence of MAPP as a coupling agent favors the formation of a strong interface [39,49].

However, further analysis of the interface fiber matrix will be given by the micromechanical analysis of the tensile strength.

Figure 3 compares the tensile properties obtained for polypropylene composites reinforced with henequen fibers and 4% MAPP with composites of the same matrix reinforced with glass fibers sized.

As shown in Figure 3a, the reinforcing capacity of glass fibers is much higher than that of henequen fibers. For a percentage of 30 wt.% of reinforcement, sized glass fibers generate an increase in tensile strength of 110% compared to 47.8% increase when henequen fibers are added. This phenomenon is mainly related to the intrinsic strength of synthetic fibers such as glass fiber, which is much higher than that of natural fibers. In this sense, glass fibers present tensile strengths of approximately 1950 MPa, carbon fibers 3950, compared to 567 MPa for sisal fibers, 400 MPa for cotton fibers or 690 MPa for hemp fibers [50,51,52]. However, higher contents of natural fibers may allow the substitution of synthetic fibers as far as tensile strength is concerned. In this regard, the tensile strength of a polypropylene composite with 20 wt.% sized glass fibers is similar to that of 50 wt.% henquen fibers (50.7 MPa vs. 52.4 MPa).

On the other hand, Figure 3b shows the elongation at break values of the composites and the corresponding value of matrix contribution to the composite strength ($\sigma_{t}^{m^{*}}$). The higher stiffness of glass fibers with Young’s Modulus between 70 and 90 GPa compared to 20–70 GPa of natural fibers such as Jute or Hemp, justifies this difference [50,52,53]. Therefore, a lower deformation capacity of the material slightly reduces the contribution of the matrix. The lower decrease in the elongation at break of the composites with henequen fibers with respect to those with glass fibers, a priori would allow further increasing the content of these fibers. However, as mentioned above, it was not technically possible to incorporate higher contents.

To compare the development of henequen fibers in the composite materials obtained versus those produced with sized glass fiber, the fiber tensile strength factor (FTSF) of the different materials was calculated, as shown in Table 3. This factor results from Equation (3), which is a rearrangement of the modified rule of mixtures (Equation (1)).
(3)$$f_{c}\cdot \sigma_{t}^{F}\cdot V^{F}=\sigma_{t}^{C}-\left(1-V^{F}\right)\cdot \sigma_{t}^{m^{*}}$$

As mentioned above, in short fiber composites, where we cannot calculate the intrinsic fiber strength experimentally, the modified rule of mixtures has two unknowns ($f_{c}$ and $\sigma_{t}^{F}$). However, by calculating the value of the product ($f_{c}\;\cdot \sigma_{t}^{F}$) for each different reinforcement volume fraction tested, we can obtain a linear correlation with origin 0 such as y = FTSF·$\;V^{F}$.

As expected, the FTSF value of the glass fiber composites is much higher than those of henequen [54]. These values agree with the results previously shown since this factor includes the intrinsic resistance of the fiber which, as indicated above, is practically three times higher than that of natural fibers. However, if the FTSFs obtained in composites reinforced with henequen fibers with and without coupling agent are compared, a notable difference is observed. Since they are the same fibers and therefore the same $\sigma_{t}^{F}$, this difference in the FTSF is, thus, attributable to the interface factor ($f_{c}$). However, these values are slightly lower than those reported in the literature, where for PP composites reinforced with wood fibers this value is above 130. These results indicate a lower intrinsic strength of non-wood fibers versus wood fibers as already reported by Ku H et al., 2011 [52].

### 3.2. Interface, Shear Strength, and Intrinsic Tensile Strength of Henequen Fibers Analysis

Micromechanical analysis of the tensile strength of henequen fiber-reinforced polypropylene composites using 4% MAPP as a coupling agent is then discussed. For this purpose, the development of the modified rule of mixtures (Equation (1)) proposed in the Kelly–Tyson model and presented in Equation (4) were used [55]. However, recently some works have been published where the behavior of natural fiber reinforced composites is modeled using other models such as the Mori-Tanaka model. This model allows the numerical and analytical homogenization of the results using Digimat software. Janowski et al. 2022 demonstrated by applying this model the dependence of the biocomposite properties on the filler geometry assumed in the calculations and on the homogenization method adopted for the calculations [56]. In addition, the results allowed the authors of the paper to estimate the usefulness of homogenization methods for predicting the properties of PHBV-hemp fiber biocomposites.
(4)$$\sigma_{t}^{C}=\chi_{1}\cdot \;\left(\sum_{i=0}^{i=l_{c}^{F}}\left[\frac{\tau \cdot l_{i}^{F}\cdot V_{i}^{F}}{d^{F}}\right]+\sum_{j=l_{c}^{F}}^{j=\infty }\left[\sigma_{t}^{F}\cdot V_{j}^{F}\cdot \left(1-\frac{\sigma_{t}^{F}\cdot d^{F}}{4\cdot \tau \cdot l_{j}^{F}}\right)\right]\right)+\left(1-V^{F}\right)\cdot \sigma_{t}^{m^{*}},$$

The main feature of the Kelly–Tyson model is that it divides the contribution of the fibers into two groups, supercritical and subcritical fibers. Fibers with a length greater than the critical length are considered supercritical and fibers with a shorter length are considered subcritical. Theoretical critical length is determined by the shear lag model (Equation (5)), which defines the critical length as that length beyond which the fibers can be fully loaded and develop their full intrinsic strength [57,58].
(5)$$l_{c}^{F}=\frac{d^{F}\;\cdot \;\sigma_{t}^{F}}{2\;\cdot \;\tau },$$

Moreover, the efficiency factor or coupling factor ($f_{c}$) presented in the modified rule of mixtures (Equation (1)) is the result of multiplying $\chi_{1}$ and $\chi_{2}$ factor $\chi_{1}$ is related to the loss of properties due to fiber orientation and $\chi_{2}$ is related to the fiber–matrix s. interface and the length of the fibers. $\chi_{1}$ takes into consideration the average angle of the fibers inside the composite to the direction in which the material is stressed. Whereas $\chi_{2}$ is a function of interface shear strength (τ), fiber aspect ratio $\left(l_{w}^{F}/d^{F}\right)$ and intrinsic fiber strength ($\sigma_{t}^{F}$) (Equation (6)).
(6)$$\chi_{2}=\tau \cdot \frac{l_{w}^{F}}{\sigma_{t}^{F}\cdot d^{F}}\;\;for\;\;l_{w\;}^{F}<l_{c}^{F},\;\chi_{2}=1-\frac{\sigma_{t}^{F\;}\cdot \;d^{F}}{4\;\cdot \;l_{w}^{F}\cdot \tau }\;\;for\;\;l_{w}^{F}\geq l_{c}^{F},$$

Kelly–Tyson model can be summarized as the fiber orientation factor by the contribution of the subcritical fibers plus the supercritical fibers to which must be added the contribution of the matrix being thus simplified as $\sigma_{t}^{C}$= $\chi_{1\;}$· (X + Y) · Z. The resolution of the Kelly–Tyson model provided by Bowyer and Bader, which considers the intrinsic tensile strength of the fibers as the product of the intrinsic Young’s modulus of the fibers ($E_{t}^{F}$) by the elongation at break of the composite material ($\varepsilon_{t}^{C}$), allows determining the four unknowns of the model $\chi_{1}$, τ, $\sigma_{t}^{F}$, and $l_{c}^{F}$ [59]. However, it is necessary to know the intrinsic modulus of the fibers which, similar to the intrinsic resistance in short fibers, is not possible to test experimentally. In this sense, by applying the Hirsch model we can obtain an approximate value for $E_{t}^{F}$ [60].

To be able to work out these mathematical models, it is necessary to determine the morphology of the fibers in the composite material. Therefore, the henequen fibers were extracted by Soxhlet extraction with Decalin solvent and analyzed using a fiber analyzer. The subsequent processes carried out at different phases of the production process, mainly extrusion in the kinetic mixer and injection, cause a significant decrease in the fiber length [6,61]. Therefore, it is crucial to know the morphology of the fibers inside the material to determine its reinforcing capacity. Table 4 shows the results of the average lengths and diameters, as well as the aspect ratio of the fibers as a function of the reinforcement content used in the preparation of the composite.

As expected, the length of the fibers extracted from the composite materials are significantly lower than those of the original fiber, which has an arithmetic length of 652 µm, a value similar to that reported in the literature [2]. As observed only in the composite with 10% reinforcement, the arithmetic length decreased by 28%. This decrease increased with the addition of a larger amount of reinforcement [62]. The decrease in fiber length can be attributed to the shear forces produced by the collision between the fibers and the walls and screw of the equipment, between the fibers and the plastic polymer, as well as the collision between fibers themselves. Substantial fiber breakage is known to occur in the early stages of processing, mainly during blending [63]. The conditions to which the fibers are subjected during this stage, such as screw speed and mixing time, are determining factors in the decrease of fiber length. However, the initial length of the fibers, their density, and hardness are also determining factors in the percentage decrease of the final length [64].

However, the diameter of the fibers is practically unchanged by the processing of the fibers in the production and transformation of the composite material. Therefore, the aspect ratio decreases with increasing fiber content in the composite material. However, if one observes the aspect ratios obtained, all of them higher than 25, it can be considered that the morphology of the fibers inside the material presents a priori enough aspect ratio to strengthen the material [65]. In previous works, it has been observed that aspect ratios higher than 20 allow effective stress transfer between the phases [47]. Small aspect ratios imply that they present a higher concentration of stresses cracking the matrix, being a point of potential failure [66]. Therefore, fiber morphology is a crucial factor in obtaining high tensile strength [67].

The results obtained from solving the Kelly–Tyson model for the different compounds produced are presented in Table 5.

The results of the micromechanical analysis show a good reinforcing capacity of henequen fibers in propylene composites. The coupling factor, except for the percentage of 10 wt.%, decreases with increasing fiber content, obtaining an average value of 0.22. The value of this coupling factor, as mentioned above, results from multiplying the orientation and length, and interface factors. Based on the works published in the literature, it is considered that a factor resulting from considering $\chi_{1}$ and $\chi_{2}$ higher than 0.18 means the achievement of a strong interface [68]. The orientation factor is mainly determined by the equipment used as well as by the shape of the injection mold and material distribution channels. In this sense, as it has been previously reported, the orientation factor in the equipment used is close to 0.3 [27,34]. As can be seen in the table, in all cases the values obtained were between 0.30 and 0.32, with an average of 0.306. From the Kelly and Tyson model, an average value of the length and interface factor of 0.74 was obtained. As in the case of the coupling factor, since $\chi_{1}$ is practically constant, the value of $\chi_{2}$ decreases as the fiber content value increases. This behavior is logical if we consider that, as shown in Table 4, the length of the fibers inside the composite material decreases with increasing fiber content. Therefore, we can consider that the decrease in $\chi_{2}$ is given by the decrease in the aspect ratio of the fibers and not by the strength of the interface. Additionally, if this value is compared with those obtained for other types of fibers, for polypropylene reinforcement, which presented a strong interface, it can be observed that the value is slightly higher, which corroborates the effect of this high aspect ratio [45]. On the other hand, for the evaluation of the interface strength, it is necessary to evaluate the shear strength value obtained; it is widely accepted that when the shear strength value is between the Tresca criterion ($\tau =\;\sigma_{t}^{m}/2$) and the von Mises criterion ($\tau \;=\;\sigma_{t}^{m}/\sqrt{3}$) [62,69]. In this sense, all shear strength values are between 13.8 and 15.9, being an average value of 14.63, therefore a strong interface can be considered.

Finally, the intrinsic strength of the fibers was determined, obtaining an average value of 398.8 MPa. This value is considered low if compared to those reported in the literature for wood fibers, however similar to those of non-wood fibers such as jute (Table 6), cotton fibers or corn stover. The intrinsic strength of the fibers with 10 and 50 wt.% reinforcement is higher than that of the other composites. In the case of 10 wt.%, the reinforcement exerted by a low fiber content is difficult to appreciate. On the other hand, in the case of 50 wt.%, it is close to the limit of content that can be incorporated, as mentioned above. It is for these two reasons that when carrying out the micromechanical analysis, the intrinsic resistance of the fibers varies, being superior to the average.

## 4. Conclusions

Henequen fibers were shown to have good reinforcement potential in thermoplastic matrices using a polypropylene-type coupling agent graphitized with maleic anhydride. Their dispersion using equipment, such as the brabender plastograhp, was good and was corroborated by the linear increase of Young’s modulus. By incorporating 50 wt.% of henequen fibers, increases in tensile strength of 89.9% were achieved, which will allow the substitution of composites with 20 wt.% of sized glass fiber. Through optical microscopy analysis, it has been possible to observe the creation of joints at the fiber–matrix interface when a coupling agent is incorporated, contrary to what occurs without this agent where it is observed that the fibers are slipped when subjected to stresses. It has been found that the need for such a coupling agent is given by a clear difference in the surface charge density of the fibers, also with 20.67 µeq/g one of the fibers with the highest surface charge among those compared. Through the micromechanical analysis, it was possible to determine an average shear strength of 14.63, a value between the Tresca and von Mises criteria. Certifying the strong interface created in the composites with a coupling agent. Finally, it was determined an intrinsic resistance of the henequen fibers of 398.8 MPa, a value similar to that of other non-wood fibers.

## Figures and Tables

**Figure 1 polymers-14-04890-f001:**
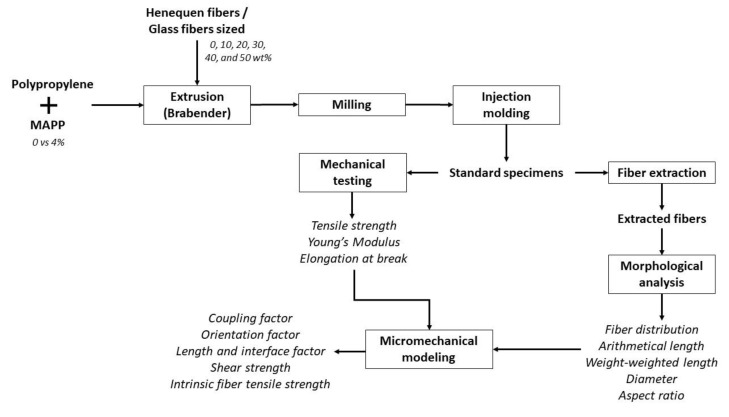
Chart-flow of composites preparation and characterization.

**Figure 2 polymers-14-04890-f002:**
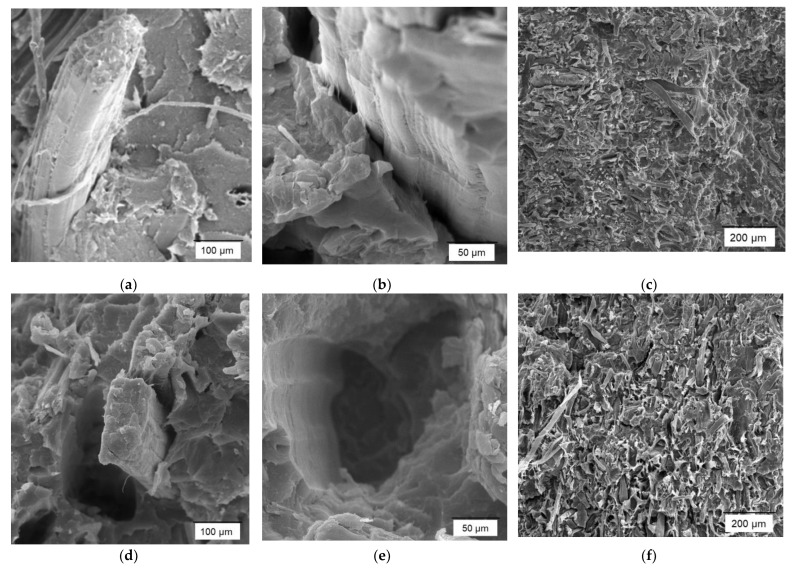
Composites reinforced with henequen fibers scanning electron microscopy images (SEM): (**a**,**b**) 30 wt.% henequen with MAPP; (**c**) 50 wt.% henequen with MAPP; (**d**,**e**) 30 wt.% henequen without MAPP; (**f**) 50 wt.% henequen without MAPP.

**Figure 3 polymers-14-04890-f003:**
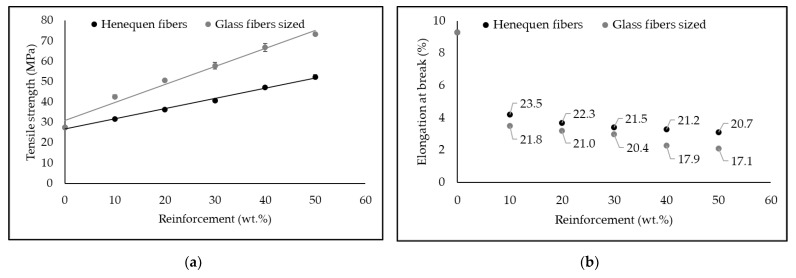
Comparison of tensile properties between Henequen fibers and glass fibers sized polypropylene composites: (**a**) Tensile strength properties of the composites; (**b**) Elongation at break of the composites and respective matrix stress at the breaking point of the composite value (data in the graph).

**Table 1 polymers-14-04890-t001:** Tensile properties for PP/Henequen composites with and without MAPP.

		Without MAPP				With 4% MAPP			
Henequen Content (wt.%)	$V^{F}$	$\sigma_{t}^{C}$ (MPa)	$E_{t}^{C}$ (GPa)	$\varepsilon_{t}^{C}$ (%)	$\sigma_{t}^{m^{*}}$ (MPa)	$\sigma_{t}^{C}$ (MPa)	$E_{t}^{C}$ (GPa)	$\varepsilon_{t}^{C}$ (%)	$\sigma_{t}^{m^{*}}$ (MPa)
0	0	27.6 ± 0.5	1.5 ± 0.1	9.3 ± 0.2	-	27.6 ± 0.5	1.5 ± 0.1	9.3 ± 0.2	-
10	0.064	27.5 ± 0.3	2.4 ± 0.1	3.8 ± 0.1	22.5	31.7 ± 0.9	2.3 ± 0.1	4.2 ± 0.2	23.5
20	0.133	29.2 ± 0.5	3.1 ± 0.1	3.5 ± 0.2	21.8	36.4 ± 0.7	3.1 ± 0.2	3.7 ± 0.1	22.3
30	0.208	30.7 ± 0.4	4.3 ± 0.2	2.5 ± 0.1	18.7	40.8 ± 1.1	4.2 ± 0.2	3.4 ± 0.2	21.5
40	0.290	35.0 ± 0.7	5.2 ± 0.1	1.9 ± 0.1	16.2	47.2 ± 1.1	5.3 ± 0.2	3.3 ± 0.2	21.2
50	0.379	38.2 ± 1.1	6.4 ± 0.2	1.3 ± 0.2	12.8	52.4 ± 1.3	6.4 ± 0.2	3.1 ± 0.1	20.7

**Table 2 polymers-14-04890-t002:** Surface charge density of PP, MAPP, Henequen fibers, and other typologies of fibers.

Material	MGCh (µeq/g)
PP	4.56
MAPP	4.67
Henequen	20.67
Jute	12.16
Sisal	23.10
Bleached Kraft pulp	8.40

**Table 3 polymers-14-04890-t003:** Fiber Tensile Strength Factor of PP composites.

Composite Material	FTSF	Ratio ^1^
Henequen/PP/without MAPP	79.82	1
Henequen/PP/with 4% MAPP	109.79	1.4
GF/PP	258.99	3.2

^1^ Ratio between the FTSF of the material and the FTSF of the henequen composite without MAPP.

**Table 4 polymers-14-04890-t004:** Morphology of the fibers extracted from the different compounds produced.

Henequen Content (wt.%)	$l_{a}^{F}$ (µm)	$l_{w}^{F}$ (µm)	$d^{F}$ (µm)	Aspect Ratio $\left(l_{w}^{F}/d^{F}\right)$
10	509.1	826.3 ± 10.6	25.5 ± 0.1	32.4
20	496.4	784.2 ± 12.3	25.6 ± 0.1	29.2
30	477.2	741.9 ± 5.9	25.3 ± 0.2	29.3
40	432.4	708.1 ± 17.2	25.4 ± 0.2	27.9
50	434.0	674.3 ± 7.2	25.5 ± 0.1	26.4

**Table 5 polymers-14-04890-t005:** Evolution of the coupling factor, orientation factor, length and interface factor, interface shear strength, and intrinsic tensile strength of henequen fibers.

Henequen Content (wt.%)	$f_{c}$	$\chi_{1}$	$\chi_{2}$	τ	$\sigma_{t}^{F}$
10	0.20	0.306	0.66	14.99	421.5
20	0.25	0.313	0.79	14.11	364.7
30	0.24	0.306	0.79	14.58	365.8
40	0.23	0.304	0.77	14.74	331.9
50	0.20	0.300	0.68	14.75	510.2
	0.22	0.306	0.74	14.63	398.8

**Table 6 polymers-14-04890-t006:** Comparison of intrinsic tensile strength values of natural fibers.

Fiber	$\sigma_{t}^{F}$	Reference
Henequen	398.8	-
Soft wood kraft	1000	[70]
Jute	306	[67]
Cotton	496	[71]
Corn stover	500	[34]

## Data Availability

Not applicable.

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
