# Peer review of "Effective Tensile Strength Estimation of Natural Fibers through Micromechanical Models: The Case of Henequen Fiber Reinforced-PP Composites"

_polymers, 2022, doi:10.3390/polym14224890_

Round 1

Reviewer 1 Report

The article "Effective Tensile Strength Estimation of Natural Fibers Through Micromechanical Models: The Case of Henequen Fibers rein-forced-PP Composites" provides a detailed account of using Henequen Fibers as an alternative for glass fibers in fabricating fiber rein-forced PP composites.  The quality of the research work presented in the article is good. Below given are some of the comments that authors should look into improving the manuscript.

  1. The authors have considered using only upto 50 wt% henequen fiber for reinforcement. How about higher percentage of using henequen?
  2. The authors have shown SEM images of composites with 30 wt% henequen fibers. But the performance of 50% was better. It would be better to show the images of the best samples so that it would be better to understand the difference in the quality of composites
  3. From the analysis of the elongation at break data, of henequen with glass fibers there is still room for using higher wt% of henequen fibers? The authors should give additional information to justify this.
  4. Authors have used 4% MAPP with the composites, how about varying the amount of MAPP, the tensile strength and quality of the composites can be improved?
  5.  Intrinsic tensile strength of 20, 30 and 40 wt% henequen fibers are lower than 10 and 50 wt%. what could be the reason for such low tensile strength.
  6. It would be better to add a comparison table showing different fiber tensile strength.

Author Response

The article "Effective Tensile Strength Estimation of Natural Fibers Through Micromechanical Models: The Case of Henequen Fibers rein-forced-PP Composites" provides a detailed account of using Henequen Fibers as an alternative for glass fibers in fabricating fiber rein-forced PP composites.  The quality of the research work presented in the article is good. Below given are some of the comments that authors should look into improving the manuscript.

The authors are grateful for the reviewer's comments, which have been used to improve the quality of the article.

  1. The authors have considered using only upto 50 wt% henequen fiber for reinforcement. How about higher percentage of using henequen?

The incorporation of henequen content of more than 50% by weight has not been technically possible. The polypropylene used is not capable of incorporating higher quantities, resulting in compounds that cannot be further processed by injection molding. The following comment has been incorporated in the section on materials and methods to clarify this fact “The composites obtained with henequen fibers have been limited to a weight percentage of 50 %, given the impossibility of polypropylene to incorporate a higher amount of fiber and maintain its cocontinuity. The incorporation in the Brabender equipment of higher contents resulted in a discontinuous material with unattached fibers, impossible to process.”. The authors are grateful for the reviewer's comment that allows to better detail the experimental procedure performed.

  1. The authors have shown SEM images of composites with 30 wt% henequen fibers. But the performance of 50% was better. It would be better to show the images of the best samples so that it would be better to understand the difference in the quality of composites

The authors are grateful for the reviewer's comment and agree with his assessment. The images are shown at 30% given the quality of the images to observe the interface. Therefore, it has been decided to maintain the detail of the interface with the 30% images and to incorporate images with 50% in Figure 2, as suggested by the reviewer.

  1. From the analysis of the elongation at break data, of henequen with glass fibers there is still room for using higher wt% of henequen fibers? The authors should give additional information to justify this.

The authors fully agree with the reviewer, the results show how considering the decrease in deformation it would be possible to incorporate a higher amount of henequen fiber. However, as mentioned above, it has not been technically possible. The following comment has been added in the results and discussion section to clarify this issue “The lower decrease in the elongation at break of the composites with henequen fibers with respect to those with glass fibers, a priori would allow further increasing the content of these fibers. However, as mentioned above, it was not technically possible to incorporate higher contents.”.

  1. Authors have used 4% MAPP with the composites, how about varying the amount of MAPP, the tensile strength and quality of the composites can be improved?

The authors agree with the reviewer that there is an optimal amount of coupling agent. In this study, the amount of 4% MAPP was used since it was optimized in a previous work. The following comment has been incorporated to clarify this point " The amount of 4% coupling agent used was optimized in previous work published by Tarrés et al. 2019 [1].".

  1.  Intrinsic tensile strength of 20, 30 and 40 wt% henequen fibers are lower than 10 and 50 wt%. what could be the reason for such low tensile strength.

The reviewer's comment is entirely pertinent. The incorporation of "limiting" contents hinders the correct evaluation of the micromechanical behavior of the composites. In the case of 10wt.%, the reinforcement exerted by a low fiber content is difficult to appreciate. In the case of 50wt.%, it is close to the limit of content that can be incorporated. It is for these two reasons that in the micromechanical analysis the intrinsic resistance of the fibers varies being superior to the average. The following comment has been incorporated in this sense "The intrinsic strength of the fibers with 10 and 50wt% reinforcement is higher than that of the other composites. In the case of 10wt.%, the reinforcement exerted by a low fiber content is difficult to appreciate. On the other hand, in the case of 50wt.%, it is close to the limit of content that can be incorporated, as mentioned above. It is for these two reasons that when carrying out the micromechanical analysis the intrinsic resistance of the fibers varies being superior to the average.".

  1. It would be better to add a comparison table showing different fiber tensile strength.

The authors are grateful for the reviewer's comment and have incorporated the table suggested by the reviewer.

Reviewer 2 Report

The work is interesting, but I have a few questions / suggestions that may extend the substantive and technical scope of the publication's content:

- Tables should not be divided, as it does not look clear enough (Table 1). In addition, the values in columns 4 and 8 do not look very clear.

- I propose to add more citations from the last five years.

-  You need to refine the publication in terms of editing.

- There is no "discussion" chapter. In my opinion it is acceptable, but in this case a deeper discussion must be undertaken when describing the results. I mean the issue of trying more to explain the observed phenomena / results and confront them with current information in the literature.

- Why were the indicated mass contents of filler in the matrix selected?

- Was the experiment planning method used?

- Natural fibers, including henequen  fibers, are characterized by a fairly wide dispersion of processing and mechanical properties. How many attempts / tests have been undertaken as part of the work carried out? Please take it into account in terms of statistics when presenting the results, including tabels (arithmetic mean, standard deviation).

- Also recently, there have been publications in the literature (eg. Materials Journal in 2022)  related to the Mechanical Properties Prediction of biocomposites with natural fibers using the DIGIMAT software using by Mori-Tanaka model or Duble Inclusion as a example of homogenization methods. This is an interesting and important issue, as natural fibers (such as henequen) are characterized by a lack of homogeneity of properties. You can write a few sentences on this topic in the discussion or in the literature introduction.

- Were examinations carried out under the microscope on the top layer of the molded samples?

Thank you in advance for preparing your answer and good luck with your publication.

Author Response

The work is interesting, but I have a few questions / suggestions that may extend the substantive and technical scope of the publication's content:

The authors are grateful for the comments made by the reviewer and have taken them into consideration to improve the quality of the article.

- Tables should not be divided, as it does not look clear enough (Table 1). In addition, the values in columns 4 and 8 do not look very clear.

The authors understand that Table 1 is particularly dense. However, they consider that it should not be divided. An attempt has been made to give more space to the table by retaining the format of the journal's template.

- I propose to add more citations from the last five years.

The authors are grateful for the reviewer's comment and have tried to incorporate more recent citations. The following citations have been incorporated:

Bhagat, A. B., & Ghosh, A. K. (2022). Performance Properties of PP/Sisal Fibre Composites having Near Critical Fibre Length and Prediction of Their Properties. Fibers and Polymers, 23(7), 1983–1994. https://doi.org/10.1007/s12221-022-4086-3

Mahesh, A., Rudresh, B. M., & Reddappa, H. N. (2022). Potential of natural fibers in the modification of mechanical behavior of polypropylene hybrid composites. Materials Today: Proceedings, 54, 131–136. https://doi.org/10.1016/j.matpr.2021.08.195

Poornima, C., Uthamballi Shivanna, M., & Sathyanarayana, S. (2022). Influence of basalt fiber and maleic anhydride on the mechanical and thermal properties of polypropylene. Polymer Composites. https://doi.org/10.1002/pc.27026

Pregi, E., Faludi, G., Kun, D., Móczó, J., & Pukánszky, B. (2022). Three-component polypropylene/lignin/flax composites with high natural additive content for structural applications. Industrial Crops and Products, 182. https://doi.org/10.1016/j.indcrop.2022.114890

Richely, E., Bourmaud, A., Placet, V., Guessasma, S., & Beaugrand, J. (2022). A critical review of the ultrastructure, mechanics and modelling of flax fibres and their defects. In Progress in Materials Science (Vol. 124). Elsevier Ltd. https://doi.org/10.1016/j.pmatsci.2021.100851

Shi, S., Yang, C., & Nie, M. (2017). Enhanced Interfacial Strength of Natural Fiber/Polypropylene Composite with Mechanical-Interlocking Interface. ACS Sustainable Chemistry and Engineering, 5(11), 10413–10420. https://doi.org/10.1021/acssuschemeng.7b02448

Janowski, G.; Frącz, W.; Bąk, Ł. The Mechanical Properties Prediction of Poly [(3-Hydroxybutyrate)-Co-(3-Hydroxyvalerate)] (PHBV) Biocomposites on a Chosen Example. Materials 2022, 15, 7531, doi:10.3390/ma15217531.

-  You need to refine the publication in terms of editing.

The authors are grateful for the comments and have tried to improve the edition according to the template.

- There is no "discussion" chapter. In my opinion it is acceptable, but in this case a deeper discussion must be undertaken when describing the results. I mean the issue of trying more to explain the observed phenomena / results and confront them with current information in the literature.

As mentioned by the reviewer, the authors intended to make only one section (Results and discussion). They have modified the section to give a greater character of discussion and comparison of the results following the reviewer's comment.

- Why were the indicated mass contents of filler in the matrix selected?

The authors have incorporated contents in the full range technically possible in fractions of 10%. The following comment has been incorporated to justify the non-addition of a higher content " The composites obtained with henequen fibers have been limited to a weight percentage of 50 %, given the impossibility of polypropylene to incorporate a higher amount of fiber and maintain its cocontinuity. The incorporation in the Brabender equipment of higher contents resulted in a discontinuous material with unattached fibers, impossible to process.".

- Was the experiment planning method used?

The experimental plan has been designed to evaluate the micromechanical tensile behavior of henequen fibers over the entire possible range of incorporation. As well as its comparison with the currently used sized glass fibers.

- Natural fibers, including henequen fibers, are characterized by a fairly wide dispersion of processing and mechanical properties. How many attempts / tests have been undertaken as part of the work carried out? Please take it into account in terms of statistics when presenting the results, including tabels (arithmetic mean, standard deviation).

The authors fully agree with the reviewer's comment. As the reviewer comments, the use of natural fibers with high disparity and presence of defects can lead to high deviations in the results. However, all tests have been performed a minimum of 10 times to obtain the arithmetic mean and standard deviation presented in tables and graphs.  It is worth mentioning that the processing of the fibers mixed in Brabender and injection molding homogenizes the fibers. The following comment has been incorporated in the materials and methods section to clarify this point " As is well known, the use of natural fibers with high disparity and presence of defects can lead to high deviations in the results. Therefore, all tests have been performed a minimum of 10 times to obtain the arithmetic mean and standard deviation presented in tables and graphs.".

- Also recently, there have been publications in the literature (eg. Materials Journal in 2022)  related to the Mechanical Properties Prediction of biocomposites with natural fibers using the DIGIMAT software using by Mori-Tanaka model or Duble Inclusion as a example of homogenization methods. This is an interesting and important issue, as natural fibers (such as henequen) are characterized by a lack of homogeneity of properties. You can write a few sentences on this topic in the discussion or in the literature introduction.

The authors are grateful for the reviewer's comment and have incorporated the following paragraph in the discussion of the results "However, recently some works have been published where the behavior of natural fiber reinforced composites is modeled using other models such as the Mori-Tanaka model. This model allows the numerical and analytical homogenization of the results using Digimat software. Janowski et al. 2022 demonstrated by applying this model the depend-ence of the biocomposite properties on the filler geometry assumed in the calculations and on the homogenization method adopted for the calculations [56]. In addition, the results allowed the authors of the paper to estimate the usefulness of homogenization methods for predicting the properties of PHBV-hemp fiber biocomposites". They also consider very interesting the incorporation of these models in their work and will study in detail their use for future work.

- Were examinations carried out under the microscope on the top layer of the molded samples?

Microscopy images were taken of the surface of the composites, but no significant differences were observed between the addition or not of coupling agent. Therefore, images were taken of the fracture where it is possible to observe the creation of holes in the absence of coupling agent.

Thank you in advance for preparing your answer and good luck with your publication

The authors would like to thank the reviewer for his good wishes as well as for the kind comments he has made. They will study in detail the Mori-Tanaka model, which will certainly improve the micromechanical analysis of the compounds.

Round 2

Reviewer 2 Report

Thank you for correcting the publication and for providing comprehensive answers to the questions. The manuscript is ready for publication.